# A Molecular Survey on Neglected *Gurltia paralysans* and *Aelurostrongylus abstrusus* Infections in Domestic Cats (*Felis catus*) from Southern Chile

**DOI:** 10.3390/pathogens10091195

**Published:** 2021-09-15

**Authors:** Natasha Barrios, Marcelo Gómez, Macarena Zanelli, Lisbeth Rojas-Barón, Paulina Sepúlveda-García, Amir Alabí, Melany Adasme, Ananda Müller, Carla Rosenfeld, César González-Lagos, Anja Taubert, Carlos Hermosilla

**Affiliations:** 1Instituto de Farmacología y Morfofisiología, Facultad de Ciencias Veterinarias, Universidad Austral de Chile, Valdivia 5090000, Chile; nbarriosveterinaria@gmail.com (N.B.); liscrb@gmail.com (L.R.-B.); 2Escuela de Medicina Veterinaria, Universidad Santo Tomas, Puerto Montt 5480000, Chile; mzanelli@santotomas.cl; 3Institute of Parasitology, Justus Liebig University Giessen, 35392 Giessen, Germany; anja.taubert@vetmed.uni-giessen.de (A.T.); carlos.r.hermosilla@vetmed.uni-giessen.de (C.H.); 4Instituto de Ciencias Clínicas Veterinarias, Facultad de Ciencias Veterinarias, Universidad Austral de Chile, Valdivia 5090000, Chile; paulina.sepulveda.garcia@gmail.com (P.S.-G.); amir_cordova@hotmail.com (A.A.); melany.adasme.bustos@gmail.com (M.A.); amuller@rossvet.edu.kn (A.M.); 5Biomedical Sciences Department, Ross University School of Veterinary Medicine, Basseterre 42123, Saint Kitts and Nevis; 6Instituto de Medicina Preventiva Veterinaria, Facultad de Ciencias Veterinarias, Universidad Austral de Chile, Valdivia 5090000, Chile; crosenfe@uach.cl; 7Departamento de Ciencias, Facultad de Artes Liberales, Universidad Adolfo Ibáñez, Santiago 8320000, Chile; cesar.glagos@gmail.com; 8Center of Applied Ecology and Sustainability (CAPES), Santiago 8331150, Chile

**Keywords:** *Gurltia paralysans*, *Aelurostrongylus abstrusus*, metastrongyloidea, domestic cat, *Felis catus*

## Abstract

*Gurltia paralysans* and *Aelurostrongylus abstrusus* are neglected metastrongyloid nematode species which infect domestic and wild cats in South American countries and in Chile, but no epidemiological studies on concomitant infections have been conducted in Chile so far. The aim of this study was not only to evaluate the occurrence of concomitant infections, but also to identify epidemiological risk factors associated with of *G. paralysans* and *A. abstrusus* infections in urban domestic cats (*Felis catus*) from Southern Chile. Blood samples from clinically healthy domestic cats from three cities of Southern Chile—Temuco, Valdivia, and Puerto Montt—were analyzed by an experimental semi-nested PCR protocol. A total of 171 apparently healthy domestic cats in Temuco (*n* = 68), Valdivia (*n* = 50), and Puerto Montt (*n* = 53) were sampled and analyzed. A total of 93 domestic cats (54.4%) were positive for *G. paralysans*, and 34 (19.9%) were positive for *A. abstrusus* infections. From those animals, 34 (19.9%) were co-infected. Cats positive with *G. paralysans* were found in all three cities; 47.2% in Puerto Montt, 48% in Valdivia, and 64.7% in Temuco. Levels of infection for *A. abstrusus* in the population under study were 4% (Valdivia), 10% (Puerto Montt), and 32.4% (Temuco). The present large-scale epidemiological study confirmed the presence of these neglected nematodes in domestic cat populations in Southern Chile, and described the possible risk factors associated with feline gurltiosis and aelurostrongylosis.

## 1. Introduction

*Gurltia paralysans* is an emerging and neglected metastrongyloid nematode (class Nematoda, order Strongylida, superfamily Metastrongyloidea, family Angiostrongylidae) occurring in domestic cats (*Felis catus*) and wild felines, such as the kodkod (*Leopardus guinia*), the northern tiger cat (*Leopardus triginus*), the margay (*Leopardus wieddi*), and the Goeffroy’s cat (*Leopardus geoffroyi*) [1,2]. The adult stages of *G. paralysans* live in the veins of the subarachnoid spinal cord and spinal cord parenchyma [1]. Pathological lesions include thrombi within the spinal veins, vein congestion, varicose veins, and consequent severe and diffuse myelopathies [1,3,4,5]. The life cycle of *G. paralysans* is currently unknown, but it has been suggested to have an indirect cycle as occurs for other closely related nematodes of the Angiostrongylidae family [1,6]. Therefore, it has been proposed that terrestrial gastropods (snails and slugs) could act as obligate intermediate hosts (IH), and multiple rodents, insects, reptiles, amphibians, and/or birds participate as paratenic hosts (PH) [6,7]. Clinical signs of feline gurltiosis normally include chronic or asymmetric ataxia of the pelvic limbs, ambulatory paraparesis, unilateral and bilateral hyperactive patellar reflexes, proprioceptive deficits, weight loss, constipation, and urinary and fecal incontinence. Laboratory alterations include mild levels of anemia, eosinophilia, and thrombocytopenia. Imaging findings by conventional myelography (CM) or computed tomographic myelography (CT-myelography), or magnetic resonance imaging (MRI), show thinning and obstruction of the dorsal and ventral spine in the thoracolumbar region, as well as a diffuse enlargement of the spinal cord in the thoracic, lumbar, and sacral regions [1,6,8]. However, despite the diagnostic tools available at present, definitive determination of feline gurltiosis can only be achieved by *post-mortem* examination revealing the presence of pre- and adult nematodes within the venous vasculature of the spinal cord [1,3]. Recent studies on diagnosis have suggested the use of a semi-nested PCR in serum samples as an additional molecular tool for allowing timely and specific detection of *G. paralysans*-derived DNA in affected cats [2,9]. 

The geographical distribution of *G. paralysans* includes areas of Chile, Argentina, Uruguay, Colombia, and Brazil, and recently it was also identified on the Island of Tenerife, Spain [1,5,10,11,12,13]. In addition to this, previous reports suggest that the focus of infection occurs mainly in the rural, urban, and peri-urban areas of Southern Chile and other South American countries [6,8]. However, no large-scale epidemiological study has been performed so far to address the occurrence of *G. paralysans* infections in domestic cat populations in Chile, or any other country. 

*Aelurostrongylus abstrusus* (class Nematoda, superfamily Metastrongyloidea, family Angiostrongylidae) is a globally distributed nematode that resides as adult stage in the lung parenchyma (bronchioles and alveolar ducts) in diverse feline host species [14,15]. Terrestrial mollusks (i.e., slugs and snails) are obligatory intermediate hosts [14,15,16,17,18,19,20]. Domestic and wild cats become infected by ingesting either IH or PH, such as small rodents, birds, amphibians, and small reptiles [16,17,18,19,20,21]. Clinical manifestations of *A. abstrusus* infection in cats vary from asymptomatic to dyspnoea, coughing, wheezing, sneezing, lethargy, mucopurulent nasal discharge, tachypnea or dyspnea, and severe bronchopneumonia, which might be fatal depending on the host immune status [14,15,17,21,22]. Young and outdoor cats are considered to be at higher risk for clinical *A. abstrusus* infections than indoor animals [14,15,16]. The gold standard diagnostic tool for feline aelurostrongylosis is the Baermann funnel method, a non-invasive and inexpensive fecal examination which relies on the detection of first-stage larvae (L1), which are infective for obligate IH [22,23]. However, it allows diagnosis only during the patency period, and the sensitivity is impaired by intermittent and/or low excretion of larvae. Molecular detection by PCR from feces or pharyngeal swabs can be used as alternative diagnostic tools for feline aelurostrongylosis [24,25,26]. An ELISA test for detection of *A. abstrusus* antibodies in serum has been used to improve the diagnosis of feline aelurostrongylosis [24,27]. Recently, a semi-nested PCR for serum samples was designed for *G. paralysans* identification and concurrent detection of *A. abstrusus* [2]. Nevertheless, this PCR protocol is still experimental for *A. abstrusus*. This molecular assay allows detection of the 28S ribosomal specific sequences of *G. paralysans* and *A. abstrusus* in serum samples, and could be used for the detection of simultaneous infections of feline gurltiosis and aelurostrongylosis [2,9]. The presence of feline aelurostongylosis in South America has been reported in Uruguay, Argentina, Brazil, Colombia, Bolivia, and Chile [28,29]. 

Although both metastrongyloid parasites have been reported in South American countries and in Chile, no studies on the levels of mono- or concomitant infections have been conducted in Southern Chile. The aim of this study was to evaluate the occurrence of *G. paralysans* and *A. abstrusus* infections in a large group of urban domestic cats from three cities (i.e., Temuco, Valdivia, and Puerto Montt) of Southern Chile, using molecular analysis. 

## 2. Results

### 2.1. Animals

A total of 171 apparently healthy domestic cats were sampled from Temuco, Valdivia, and Puerto Montt from January 2019 to December 2019. The number of domestic cats from each city was as follows: Temuco, *n* = 68; Valdivia, *n* = 50; and Puerto Montt, *n* = 53 (Table 1). Some information requested by the questionnaire was unfortunately not provided by the owners. Epidemiological and individual data are reported in Table 2. From the total domestic cat population, 45% (*n* = 77) were female and 40.4% (*n* = 69) were male; in 14.6% (*n* = 25) this information was not registered. The age of the study animals ranged between 4 and 132 months old (average 32 months). The animals had either indoor (16.4%) or outdoor (37.4%) access, but for 46.2% the information was not provided by the owners. Anthelmintic treatments, according to owners, were performed in 22.2% of analyzed domestic cats (Table 2). A total of 118 animals (69%) were mentioned to live exclusively in urban settlements and only 3 individuals (1.8%) in rural settlements as well; 50 owners did not respond to this question. 

### 2.2. Descriptive Epidemiology

From the population under study, a total of 93 cats (54.4%) were positive for *G. paralysans* infections, and 34 (19.9%) for *A. abstrusus* infections. Co-infections of *G. paralysans* and *A. abstrusus* were detected in 34 domestic cats (19.9%). The distribution of *G. paralysans* and *A. abstrusus* infection in the different cities is shown in Table 1 and Figure 1. Cats positive with *G. paralysans* found in the three cities ranged from 47.2% (Puerto Montt), 48% in Valdivia, and 64.7% in Temuco. The rate of infection percentage for *A. abstrusus* in the analyzed cat population under study ranged from 4% (Valdivia) and 10% (Puerto Montt) to 32.4% in Temuco. 

### 2.3. Epidemiological Factors

The results of the present study showed a lack of association between the analyzed potential risk factors and risk of infection for *A. abstrusus* and *G. paralysans* (Table 3 and Table 4).

### 2.4. Molecular Detection of G. paralysans and A. abstrusus

The concentrations of DNA extracted from the feline blood samples ranged between 12.52 ng/µL and 172.48 ng/µL, with purity indices higher than 1.4, which guaranteed their viability and, therefore, their optimal amplification, and allowed them to be considered completely intact DNA to be used in subsequent species-specific PCR reactions. Consequently, molecular analysis was carried out through conventional PCR, which amplified a common metastrongyloid sequence of approximately 450 bp (Appendix A).

After the general tests, the species-specific semi-nested PCR assay allowed us to determine the presence of *G. paralysans*, even in those samples where amplification of the common metastrongyloid fragment was not achieved. In this way, the positivity index could be evaluated in the three cities in Southern Chile. The amplified fragment in the semi-nested reactions consisted of 356 and 300 bp for *G. paralysans* and *A. abstrusus,* respectively (Appendix A).

Sequencing analysis in seven aleatory samples was performed in the AUSTRAL-omics Core Facility (Faculty of Sciences, Austral University of Chile), and confirmed the identification of 18S ribosomal RNA gene fragments with high identity with *G. paralysans* (GenBank Accession number JX975484). Additionally, two aleatory samples confirmed the identification of the 28S rRNA gene for *A. abstrusus* (GenBank Accession number AM039759.1) (Appendix A).

## 3. Discussion

This study constitutes the first large-scale molecular survey on the presence of *G. paralysans,* as well as *A. abstrusus* infection, in clinically healthy domestic cats from Chile. The overall prevalence of *G. paralysans*-positive urban cats in this study from three cities of Southern Chile was 54.4% (93/171). The level of infection among the three evaluated cities varied between 48 and 64%. No large-scale epidemiological studies on the prevalence of this neglected feline nematode species have been conducted previously, either in Chile or elsewhere worldwide. However, several studies in Chile have reported clinical cases of feline gurltiosis in areas of Southern Chile, including Lastarria (the La Araucania region), Punucapa, Niebla, Paillaco, and Futrono (the Los Ríos region), and Encenada, Lago Ranco, Pichirropulli, and Ancúd (the Los Lagos region) [1,3,9]. In the rest of the South American continent, feline gurltiosis cases have been reported in the provinces of Antioquia, Colombia, Baradero (Buenos Aires province), Las Colonias, Santa Fé province in Argentina, Fray Bentos in Uruguay, and Rio Grande do Soul and Pernambuco in Brazil [10,11,12,30,31]. Epidemiological studies on closely related canine lungworms (*Angiostrongylus vasorum*, *Crenosoma vulpis*) have noted a high risk of metastrongyloid nematode infection in South America based on climatic variables and their effects on the survival rates of possible obligate IH (terrestrial or aquatic mollusk) [24]. Outside the Americas, one report of feline gurltiosis from Macaronesian Canary Island in Spain has recently been published, thereby expanding the geographic endemicity of *G. paralysans* to Europe [13]. The presence of *G. paralysans* in Tenerife Island could be due to the introduction of *G. paralysans*-infected domestic cats from endemic areas of South America, or to the importation of either infected IH or PH [1,13]. 

Conversely, *A. abstrusus* infections have been reported in several countries in Europe, North America, and South America [29,32,33,34,35,36,37]. The overall occurrence of *A. abstrusus* in our molecular study was 19.9% (34/171), and the prevalence values between the three cities ranged from 4% to 32%. The difference among the levels of infections of *A. abstrusus* observed in this study may reflect the level of transmission or availability of IH and PH that are able to maintain the lifecycle of this nematode in certain areas. In another study, within the city of Valdivia, a prevalence of 38% was determined by bronchial lavage and *post-mortem* examination; both methods are considered the most sensitive tools for a final diagnosis, rather than coprological evaluation [38]. In another study in the cities of Río Bueno and La Unión (Los Ríos region), also in the southern part of Chile, the level of infection, determined by Baermann funnel coprological examination, was 10% (20/200) [39]. The prevalence of *A. abstrusus* infections in other South American countries has varied between 0.21 to 39% [20,21]. In Brazil, an occurrence of 29.5% (24/88) and 39% (40/102) was observed in two studies in Rio Grande do Sul, 18.6% (range 5.9–25%) in Santa María, 18% in Uberlândia, 8.5% in Sao Paulo, and 1.3% in Cuiba and Várzea Grande, Matto Grosso [40,41,42,43]. Studies in wild felids (e.g., *L. wieddii*, *L. triginus*) in Brazil have reported prevalence rates of 38.1% and 35.7% for *A. abstrusus* infections, respectively [44]. Interestingly, recent studies have reported *G. paralysans* infections in both wild cat species in Brazil as well [45,46]. In Montevideo, Uruguay, a prevalence of 8.6% for *A. abstrusus* was estimated by *post-mortem* examination in domestic cats [47]. In Argentina, studies have determined a prevalence of 24.3% in La Plata, 30% in Corrientes [48,49], and 35.3% (6/17) in Buenos Aires [50]. In Colombia, studies have estimated a prevalence of 0.21% (1/121) in the Quindío, and 0.4% (2/473) in Antioquia [28]. In line with these findings, feline aelurostrongylosis has been diagnosed throughout Europe, including Germany, France, Italy, Spain, Portugal, Albania, Croatia, Greece, Turkey, Israel, Denmark, England, Rumania, Austria, and Belgium, with prevalence rates between 0.3% and 50% depending on the region, lifestyle, and diagnostic methods (e.g., coprological, serological, or molecular methods) [33,34,35,36,37,51,52,53,54,55]. In the USA, prevalence rates of 2.07% in a retrospective study, 6.2% in New York, and 18.5% in Alabama have been reported in shelter and stray cats [56,57,58]. 

PCR-based diagnostic methods have been used in the past for alternative diagnosis of *A. abstrusus* in fecal samples, by amplification and analysis of ITS-2 sequences [59]. Additionally, concurrent detection and differentiation of *Troglostrongylus brevior* and *A. abstrusus*, also in fecal samples, based on ITS-2 duplex PCR assay, has been previously reported [25]. Molecular studies can overcome the limitations of coprological analysis, which may underestimate the prevalence of parasite infections during pre-patent or post-patent periods, and thus not detect all infected animals [25]. In fact, previous studies have shown that some cats gave false negatives for *A. abstrusus* on copromicroscopic evaluation, and were confirmed positive by nested-PCR on pharyngeal swabs [60]. 

In the current epidemiological study, co-infections of *A. abstrusus* and *G. paralysans* were observed in 19.9% of animals, with a range of 4 to 18.9% in the three investigated cities. Interestingly, all *A. abstrusus* positive cats were also positive for *G. paralysans*. Previous *post-mortem* studies in domestic cats in Chile suffering feline gurltiosis also showed the presence of *A. abstrusus* adult and larval stages in the pulmonary tissues [8]. This co-infection may indicate that both metastrongyloid parasites might share the same IH or PH as postulated elsewhere [1,9]. Suitable obligate IH for *A. abstrusus* in South America are considered the terrestrial mollusk species *Rumina decollate* or the neozoan African giant snail *Achatina fulica*, which should be investigated for the presence of *G. paralysans* infective third-stage larvae (L3) in future monitoring studies on gastropod-borne lungworms, as recently performed in Colombia [29] and in the Macaronesian Archipelago of Spain [33], where feline gurltiosis has recently been documented [13]. Additionally, these new data suggest that the southern region of Chile may offer suitable biological and epidemiological conditions for the occurrence and spread of these neglected metastrongyloid feline nematodes. 

The conducted multivariate analysis showed that neither individual nor environmental factors were statistically significant in relation to the occurrence of mono or concomitant *A. abstrusus*/*G. paralysans* infections. Frequency of prey captured by the domestic cats, according to owners in this study indicated that rodents and birds were the most commonly hunted prey species. In a recent study in Chile concerning hunting behavior of owned cats, most cat owners (84.1%) reported that birds were the most common type of prey species (49.9%), followed by diverse rodent species (39.3%), insects (29.5%), lizards (20.2%), rabbits (0.95), and even bats (80.4%) [61]. Considering the fact that terrestrial snails and/or slugs are hunted by felines to a lesser extent than other prey species, it seems reasonable to investigate several PH for *A. abstrusus* (and probably for *G. paralysans*), including scavenger insects, birds, reptiles, amphibians, and small mammals, to better understand the complex epizootiology of feline gurltiosis [62]. The high level of infection with *G. paralysans* and *A. abstrusus* in the current study could indicate that appropriate niches occur in these areas that favor the infection of both nematodes. Of note, all geographic areas surrounding the cities of Temuco, Valdivia, and Puerto Montt correspond to ecosystems of the Valdivian rainforest, characterized by an ecoregion of Central-Southern Chile, with temperate rainfalls, an oceanic mild climate, abundant gastropod IH and PH species, other potential definitive feline hosts (i.e., kodkods, pumas (*Puma concolor*)), and abundant forested areas [1]. 

In the present study, domestic cats under one year of age showed higher prevalence rates for *G. paralysans* and *A. abstrusus* infections than older animals. Studies in Northern and Central Italy with more than 800 domestic cats indicated that cats younger than one year were at a higher risk of infection with *A. abstrusus* and *T. brevior* [63,64]. Another study in Denmark reported a lower risk of *A. abstrusus* infection in kittens younger than 11 weeks, compared to older cats [54]. However, other studies have reported that age, as well as gender, do not seem to be risk factors for *A. abstrusus* infections [65,66].

Limitations of the study included a lack of copromicroscopic methods (i.e., Baermann technique, fecal smears and/or fecal flotations) for the detection of L1. Although the semi-nested PCR technique used in our study has been validated for *intra vitam* diagnosis of *G. paralysans* using serum samples, further validation is required for *A. abstrusus* detection [2,9]. However, positive *A. abstrusus* DNA samples in our analysis were confirmed with posterior sequencing analysis for the 28S rRNA gene. The molecular analysis used in this study could be an alternative method for the diagnosis of mixed feline infections with *G. paralysans* and *A. abstrusus* as previously suggested [9]. As already stated, further work is required for validation of this molecular diagnostic tool for aelurostrongylosis detection in blood samples, using simultaneous Baermann funnel assays and classical PCR protocols. 

Thus, these results should be interpreted with caution, given that this cat population was not sampled aleatorily, making estimation of the general prevalence of both nematodes difficult to stablish in Southern Chile. Nonetheless, our results clearly indicate that *G. paralysans* and *A. abstrusus*-infections are endemic in urban areas, with rather high prevalence rates particularly for feline gurltiosis. Molecular diagnostic techniques will hopefully support increased awareness and knowledge of the epidemiology and geographical distribution of these two underestimated feline metastrongyloid parasitoses in the future, not only in Chile, but also in other South American and European countries. 

## 4. Materials and Methods

### 4.1. Animals and Study Area

The domestic cats originated from three cities of Southern Chile: Temuco (La Araucania region, 38° S 73° W), Valdivia (Los Ríos region, 39° S 73° W), and Puerto Montt (Los Lagos region, 41° S 73° W) (Figure 1). All study areas were located in Southern Chile, where previous cases of feline gurltiosis had been reported. Blood samples were taken from different Veterinary Hospitals from the cities of Temuco, Valdivia, and Puerto Montt, Chile, which voluntarily collaborated with this study. The cats were sampled regardless of age and gender. For this study, domestic cats without manifestations of neurological signs, or that were treated for problems other than neurological ones, were selected. Data obtained at the time of presentation included age, gender, lifestyle (i.e., indoor/outdoor activities), and anthelmintic treatment, for potential risk factors analysis. Each owner signed an informed consent form before sampling of the animals. 

### 4.2. Selection of Participants and Biological Samples

Blood samples were collected individually from 171 domestic cats for which informed consent had been obtained from their owners. The animals came from the cities of Valdivia (*n* = 50), Puerto Montt (*n* = 53), and Temuco (*n* = 68), Southern Chile (Figure 1). Blood samples were collected from January 2019 to December 2019. Blood was collected from the cephalic or external jugular vein of each animal using tubes containing acid citrate dextrose anticoagulant (ACD) (Vacutainer, BD Biosciences, Franklin Lakes, NJ, USA). The blood samples were conserved on wet ice in an insulated container, then transported to the Austral University of Chile (UACh), Clinical Pathology Laboratory, and stored at –80 °C until required. 

### 4.3. Extraction and Quantification of G. paralysans and A. abstrusus–Specific DNA

DNA extraction from the serum samples was performed according to the manufacturers instructions, using an E.Z.N.A.^®^ Tissue DNA Kit D3396-02 (Omega Bio-tek, Inc., Norcross, GA, USA). Subsequently, the DNA samples were quantified by spectrophotometry (Thermo Scientific NANODROP 2000, Waltham, MA, USA) and stored at −20 °C until use. Molecular detection of *G. paralysans* was by semi-nested PCR. Two PCR assays were performed, one general and two species-specific, for the detection of DNA from *G. paralysans* and *A. abstrusus* [9]. The first general PCR assay amplified a common DNA sequence of the metastrongyloids using the universal primers U1 (AaGp28Sa1) and U2 (AaGp28Ss1). The second specific test consisted of two semi-nested PCRs, which differentiated between DNA from *G. paralysans* and *A. abstrusus* with a mixture of universal and specific primers; in this case, U1/E2 (Gp28Sa3) for *G. paralysans* and U2/E1 (Aa28Ss2) for *A. abstrusus*. This last species is phylogenetically very close to *G. paralysans*. The specific primers were designed from the multiple alignments of the 28S subunit of the metastrongyloid rDNA (Table 5).

The reaction mix for the general and specific PCR assays consisted of GoTaq Green Master Mix 2X^®^ Polymerase (Promega Corporation, Fitchburg, MA, USA), with a final concentration of 1X, 10 µM of each primer (general or specific assay), approximately 200 ng of DNA obtained from the blood samples, and a sufficient quantity of nuclease-free sterile water for a final volume of 25 µL. Semi-nested PCR assays were performed on all amplicons from the general assay, regardless of whether they had been positive in the first reaction or general reaction. The positive control used in the molecular analysis was DNA samples positive for feline gurltiosis in previous investigations. The negative control consisted of the same reaction mixture with nuclease-free water and without DNA.

The amplification for the general PCR assay was adapted to the following conditions: initial denaturation at 94 °C for 5 min, followed by 35 denaturation cycles at 94 °C for 30 s, alignment at 54 °C for 30 s, extension at 72 °C for 30 s, and a final extension at 72 °C for 5 min. Similarly, the amplification conditions established for the semi-nested PCR assays consisted of an initial denaturation step at 94 °C for 4 min, followed by 35 denaturation cycles at 94 °C for 30 s, alignment at 55 °C for 1.5 min, extension at 72 °C for 30 s, and the final extension at 72 °C for 4 min.

The PCR products were analyzed by electrophoresis on 2% agarose gels (Fermelo BIOTEC, Santiago, Chile), prepared in TAE (Tris-Glacial Acetic Acid-EDTA) buffer, supplemented with 0.01% intercalating staining agent SYBR^®^ Safe DNA Stain (Invitrogen, Paisley, UK), and developed in the presence of ultraviolet light. A 100 bp molecular weight marker (Maestrogen, Hsinchu, Taiwan) was implemented. The established electrophoresis conditions were 90 V, 100 mA, and 10 W for one hour.

### 4.4. Amplification of IRBP as an Internal Control by Conventional PCR

The IRBP (interphotoreceptor retinoid binding) gene was amplified as a specific internal control for vertebrate mammals [67]. The PCR reaction mixture consisted of GoTaq Master Mix 2X (Waltham, MA, USA), 10 µM of each forward and reverse primer, approximately 200 ng of DNA obtained from the blood samples, and a sufficient quantity of nuclease-free sterile water for a final volume of 10 µL. The amplification conditions consisted of an initial denaturation at 94 °C for 4 min, followed by 35 cycles of denaturation at 94 °C for 30 s, alignment at 57 °C for 30 s, extension at 72 °C for 30 s, and a final extension at 72 °C for 5 min. The electrophoresis analysis was carried out under the same conditions as previously described.

### 4.5. Sequencing Analysis

Seven DNA samples that were previously positive for *G. paralysans* and two positives for *A. abstrusus* were randomly selected for sequencing analysis. The purification of the PCR products was carried out following the manufacturer’s instructions, using the Silica Bead DNA Gel Extraction^®^ kit (Thermo Scientific, Waltham, MA, USA). The results obtained were evaluated using the biological sequence editing software BioEdit Version 7.0.9.0, and the Basic Local Alignment Search Tool (BLAST) available at NCBI (National Center for Biotechnology Information).

### 4.6. Questionnaire for Domestic Cat Owners

A questionnaire form was provided to all domestic cat owners, which gathered information concerning the origin of the cat, gender, age, indoor/outdoor lifestyle, anthelmintic treatments, and hunting behavior (including the type of prey species).

### 4.7. Data Analysis

Molecular positivity for metastrongyloid nematodes and coinfection was summarized by relative frequencies and contingency tables. To assess for statistical significance of the potential risk factors associated with infection with *G. paralysans, A. abstrusus* and their coinfection (‘positive’ versus ‘negative’ as the response variable), we estimated mixed effects logistic regression models. Age (in months), sex (‘male’ versus ‘female’), and hunting behavior (‘yes’ versus ‘no’) were used as individual level predictors. Anthelmintic use (‘yes’ versus ‘no’) and lifestyle (‘indoor’ versus ‘outdoor’) were used as management level categorical predictors. Hunted species were not considered because of the low sample size (25% of the total cats, from two cities). A random intercept by city of origin was evaluated (i.e., cats originating from the same city may share epidemiological and environmental factors), but this factor produced model overfitting, and thus was not included in the models. Risk factors, expressed as an odds ratio (OR) with a 95% confidence interval, were determined by univariate and multivariable logistic regression analysis, where a *p*-value of ≤ 0.05 was considered statistically significant. These analyses were performed with the package “lme4” implemented in the R software (R Core Team 2017).

## 5. Conclusions

In conclusion, our results demonstrated occurrence of *G. paralysans* and *A. abstrusus* in clinically unaffected urban domestic cat populations of three Southern Chilean cities. Additionally, the results indicated that infection levels of *G. paralysans* and *A. abstrusus* seem to vary according to the geographical region in Southern Chile. Co-infection of these metastrongyloids occurred in a considerable percentage of analyzed domestic cats. Therefore, we encourage future studies on epizootiological drivers of feline gurltiosis and aelurostrongylosis occurring in urbanized, peri-urban, and rural/wild geographic areas.

## Figures and Tables

**Figure 1 pathogens-10-01195-f001:**
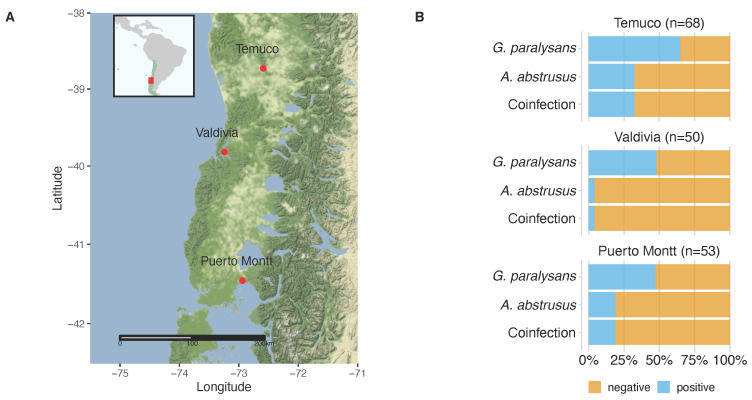
Study location and molecular positivity of feline gurltiosis and aelurostrongylosis in domestic cats from Southern Chile. (**A**) Geographical localization of the three cities in Southern Chile, where 171 domestic cats were analyzed for infections of *Gurltia paralysans* and *Aelurostrongylus abstrusus*. The small box contains a map from Latin America; Chile is highlighted by green color and the study area by a red rectangle. The main map contains the study area and the cities are represented by red points and names. (**B**) Proportion of molecular positivity obtained by semi-nested PCR of 171 domestic cats for *Gurltia paralysans*, *Aelurostrongylus abstrusus**,* and co-infections.

**Table 1 pathogens-10-01195-t001:** Molecular positivity obtained by semi-nested PCR of 171 domestic cats for Gurltia paralysans and Aelurostrongylus abstrusus in the cities of Temuco, Valdivia, and Puerto Montt, Southern Chile.

	*G. paralysans*		*A. abstrusus*		*A. abstrusus* + *G. paralysans*	
City	Positive/Total	%	Positive/Total	%	Positive/Total	%
Puerto Montt	25/53	47.2	10/53	18.9	10/53	18.9
Temuco	44/68	64.7	22/68	32.4	22/68	32.4
Valdivia	24/50	48	2/50	4.0	2/50	4.0
Total	93/171	54.4	34/171	19.9	34/171	19.9

**Table 2 pathogens-10-01195-t002:** Frequency of demographic features collected from 171 domestic cats analyzed for Gurltia paralysans and Aelurostrongylus abstrusus infections in Southern Chile.

*G. paralysans*			*A. abstrusus*	
	Positive/Total	%	Positive/Total	%
**Gender**				
female	41/77	46.8	15/77	19.5
male	37/69	46.4	19/69	27.5
n/d	15/25	40.0		
Total	93/171	45.6	34/146	23.3
**Indoor/outdoor**				
indoor	12/28	57.1	14/64	21.9
outdoor	33/64	48.4	7/8	25.0
n/d	48/79	39.2	13/79	16.5
Total	93/171	45.6	34/171	19.9
**Anthelmintic**				
no	23/55	58.2	8/55	14.5
yes	23/38	39.5	13/38	34.2
n/d	47/78	39.7	13/78	16.7
Total	93/171	45.6	34/171	19.9
**Hunting**				
no	22/50	56.0	13/50	19.0
yes	23/42	45.2	8/42	26.0
n/d	48/79	39.2	13/79	16.5
Total	93/171	45.6	34/171	19.9
**Age**				
<1 year	24/43	44.2	8/43	18.6
≥1 year	49/95	48.4	23/95	24.2
n/d	20/33	39.4	3/33	9.1
Total	93/171	45.6	34/171	19.9

n/d = no data available.

**Table 3 pathogens-10-01195-t003:** Molecular positivity of metastrongyloids in the context of potential risk factors associated with infection. Results are shown from a generalized lineal model accounting for variation in the molecular positivity of *Aelurostrongylus abstrusus* as a function of age (adult versus young), sex (female versus male), lifestyle (indoor versus outdoor), anthelmintic use (no versus yes), and hunting behavior (no versus yes). The category in brackets is compared to the reference factor.

*A. abstrusus*			
Predictors	Odds Ratios	CI	p
age [young]	1.24	0.36–4.08	0.72
sex [male]	1.01	0.35–2.91	0.98
lifestyle [outdoor]	1.88	0.47–8.03	0.37
anthelmintic [yes]	3.92	1.33–12.77	0.02
hunting [yes]	0.40	0.10–1.59	0.19

CI: confidence interval.

**Table 4 pathogens-10-01195-t004:** Molecular positivity of metastrongyloids in the context of potential risk factors associated with infection. Results shown are from a generalized lineal model accounting for variation in the molecular positivity of *Gurltia paralysans* as a function of age (adult versus young), sex (female versus male), lifestyle (indoor versus outdoor), anthelmintic use (no versus yes), and hunting behavior (no versus yes). The category in brackets is compared to the reference factor.

*G. paralysans*			
Predictors	Odds Ratios	CI	p
age [young]	1.55	0.57–4.29	0.39
sex [male]	1.28	0.53–3.10	0.58
lifestyle [outdoor]	1.64	0.52–5.39	0.40
anthelmintic [yes]	2.36	0.96–6.03	0.06
hunting [yes]	1.02	0.34–3.07	0.96

CI: confidence interval.

**Table 5 pathogens-10-01195-t005:** Universal primers and specific and internal controls used in the molecular detection of *Gurltia paralysans* and *Aelurostrongylus abstrusus* DNA.

Oligonucleotide	Target	Primer Name	Sequence
Universal	Metastrongyloidea	AaGp28Ss1	5’-CGAGTRATATGTATGCCATT-3´
Universal	Metastrongyloidea	AaGp28Sa1	5´-AGGCATAGTTCACCATCT-3´
Specific	*Gurltia paralysans*	Gp28Sa3	5´-TCTTGCCGCCATTATAGTAG-3´
Specific	*Aelurostrongylus abstrusus*	Aa28Ss2	5´-CGTTGATGTTGATGAGTATC-3´

## Data Availability

Not applicable.

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
