# Peer review of "A Molecular Survey on Neglected Gurltia paralysans and Aelurostrongylus abstrusus Infections in Domestic Cats (Felis catus) from Southern Chile"

_pathogens, 2021, doi:10.3390/pathogens10091195_

Round 1

Reviewer 1 Report

Dear authors,

the topic of the article is of sound interest in Veterinary Parasitology and there is the merit in gaining new and fresh knowledge on feline metastrongyloids and on the diagnostic techniques, particularly on G. paralysans that is very important since it is emergent and neglected. Also, molecular information on A. abstrusus and on lungworms in general in South America is very poor.

Nevertheless, I am afraid that this work has several weaknesses that should mandatorily be addressed before setting up any discussion and before your results can be considered for publication. Moreover, the manuscript requires extensive and careful English revision, typo checking and a complete revision of the reference list, matching text/references, and an update of the literature.

Please find below Major and Minor comments.

Major comments:

1) Although the authors state that the lack of use of copromicroscopic techniques in their work is a limitation, they must clearly specify why they choose to use PCR on blood samples for the diagnosis of A. abstrusus infection. Indeed, it is very rare to see epidemiological studies based on this technique. This should be clearly mentioned before putting in place any kind of discussion (in the m&m section).

The most sensitive tools for the detection of A. abstrusus in epidemiological studies is represented by the Baermann method; serological studies are useful to evaluate the level of exposure in given populations. These two methods could benefit from a molecular confirmation. This is clearly stated in the paper that the author cite (it should be Raue et al., 2020 - hard to say due to the wrong renumbering of the references), when they refer to the pcr on blood, but Raue et al., 2020 does not mention any PCR on blood samples.

This would be the first time that an epidemiological study is based only on the use of blood-based PCR.

Therefore, it is this reviewer opinion that it is mandatory for this kind of article (and of course it would be of sound interest) to explain:

- the rationale of the choice of using only PCR on blood for A. abstrusus

- what are the sensitivity and sensibility of PCR on blood for A. abstrusus

- how the authors intend to compare a blood-based PCR for A. abstrusus to the conventional PCR used in the vast majority of epidemiological studies, that is performed on faeces (Baermann sediment) or pharyngeal mucous for A. abstrusus

- if and how the authors standardized the method they used.

- how they explain the positivity to A. abstrusus at PCR of blood samples (and they should also reference this) 

In the method section, the authors indeed refer only to a paper on the molecular detection of Gurltia, but not of Aeluro. Indeed, the sensitivity/specificity of this technique in finding A. abstrusus DNA in the blood of cats is currently unknown.

This primary major concern should be addressed before rendering this paper publishable.

The authors should also include somewhere in the introduction a clear explanation of the rationale for the usage of this technique, other than pointing out advantages and disadvantages in the discussion section, otherwise their results cannot be considered for publication.

2) The abstract is 340 and should be drastically reduced as a maximum of 200 words is indicated for the abstract in the authors guidelines. 

3) The manuscript requires an extensive and careful English grammar check, as well as a check of the large number of typos contained in it. Also, the manuscript is redundant and the language should be simplified

4) The reference list should be checked very carefully, as well as the matching text/references throughout the text, that seem to be completely wrong.

For instance, refs 8 and 9 are put in relation with sentences on Gurltia, while these refs refer actually to A. abstrusus. This makes the verification of the information extremely difficult. As far as I understood by reading the reference list: the refs in the text from 6 on, should be considered with 6 numbers more in the reference list. For instance, as far as I understood the n.8 in the reference list corresponds to nr. 14 in the text (I think).

In any case, regarding A. abstrusus the authors refer to fairly old articles, i.e. Traversa et al., 2008 for biological aspects and Elsheikha et al., 2016, of 13 (really old) and 5 years ago respectively.

It is here strongly recommended to cite updated literature here, as new knowledge has been gained in the last years and reviewed in other very recent papers published on Pathogens, e.g.

https://pubmed.ncbi.nlm.nih.gov/33401704/ (generalist/epidemiological/biological review on feline lungworms)

https://pubmed.ncbi.nlm.nih.gov/33920104/ (review on the clinical aspects)

5) The discussion section is quite inconclusive, and it is only partially differentiate between the techniques used in the different studies cited when comparing prevalences. Moreover, discussions should be reformulated basing on the major issues raised on the methodology used. A complete and extended revision of English is strongly recommended, as well as a careful rephrasing as information is difficult to understand in the present form. Updated literature should be cited.

MINOR COMMENTS

Lines 54-55: This sentence is redundant and it is not clear what the authors mean for “angio and neuotropic tropism”. Please rephrase

Line 57: replace “this parasite” with “G. paralysans

Lines 61-69: The authors should be careful when writing “the diagnosis of gurltiosis consists basically of”. Indeed, all this part should be preferabily presented as:

Clinical signs of feline gurltiosis

Laboratory alterations

Imaging diagnosis tools

Indeed, the authors themselves thereafter state that a diagnosis is achievable only post-mortem

 Line 72: Please check “pre-and adult the adult”.

Lines 73-74 It is not clear in which circumstances and which biological samples should be subjected to PCR for a diagnosis of G. paraylisans infections. Please specify4

Line 83-85 Please cite that different species of gastropods are known to act as IH for A. abstrusus, as shown in both experimental and field studies:

https://pubmed.ncbi.nlm.nih.gov/31002674/

https://pubmed.ncbi.nlm.nih.gov/32650821/

https://www.mdpi.com/2076-0817/10/8/960

https://pubmed.ncbi.nlm.nih.gov/33322102/

Line 118-151 Scientific names are not in Italics here. Please change

Line 94 it is swabs not swaps. Please refer to updated literature on the diagnostic tools. Indeed, to the best of this reviewer’s knowledge PCR on blood samples has very little (if not nil) utility in supporting the diagnosis of feline aelurostrongylosis (major comments)

Line 95-98 The authors propose a molecular based study, and this is quite innovative other than very useful for studies investigating lungworms in South America, as recently proposed in recent paper on lungworms in Brazil: https://pubmed.ncbi.nlm.nih.gov/34068219/

Line 108: questionnaire

Line 255 maybe the authors meant “clinically healthy” cats

Line 260 “report.. have reported”. Please delete redundancies throughout the text

Line 270 suppose it “the” here and not “de”. It would be of benefit to provide explanations for the finding in Tenerife.. maybe is it due to increased animal movimentations or to a lack of awareness? Please specify and reference

Line 285 determined

Line 301 the list of references regarding Europe should be updated. These are only examples of very recent studies:

https://pubmed.ncbi.nlm.nih.gov/30857861/

https://pubmed.ncbi.nlm.nih.gov/32748040/ (different techniques)

https://pubmed.ncbi.nlm.nih.gov/31991881/ (molecular study - faeces)

https://pubmed.ncbi.nlm.nih.gov/30552978/

The authors should explain why they decided to use a PCR on blood for the detection of A. abstrusus, since this method is not the best option for its diagnosis. Moreover, its sensitivity/specificity compared to best-known techniques (e.g. Baermann method, serology, PCR on other biological samples such as Baermann sediment or pharyngeal swabs - mucus) is unknown.

Author Response

RESPONSE TO REVIEWER 1

Dear authors,

the topic of the article is of sound interest in Veterinary Parasitology and there is the merit in gaining new and fresh knowledge on feline metastrongyloids and on the diagnostic techniques, particularly on G. paralysans that is very important since it is emergent and neglected. Also, molecular information on A. abstrusus and on lungworms in general in South America is very poor.

Nevertheless, I am afraid that this work has several weaknesses that should mandatorily be addressed before setting up any discussion and before your results can be considered for publication. Moreover, the manuscript requires extensive and careful English revision, typo checking and a complete revision of the reference list, matching text/references, and an update of the literature.

Please find below Major and Minor comments.

Major comments:

Although the authors state that the lack of use of copromicroscopic techniques in their work is a limitation, they must clearly specify why they choose to use PCR on blood samples for the diagnosis of A. abstrusus infection. Indeed, it is very rare to see epidemiological studies based on this technique. This should be clearly mentioned before putting in place any kind of discussion (in the m&m section).

The most sensitive tools for the detection of A. abstrusus in epidemiological studies is represented by the Baermann method; serological studies are useful to evaluate the level of exposure in given populations. These two methods could benefit from a molecular confirmation. This is clearly stated in the paper that the author cite (it should be Raue et al., 2020 - hard to say due to the wrong renumbering of the references), when they refer to the pcr on blood, but Raue et al., 2020 does not mention any PCR on blood samples.

This would be the first time that an epidemiological study is based only on the use of blood-based PCR.

Therefore, it is this reviewer opinion that it is mandatory for this kind of article (and of course it would be of sound interest) to explain:

- the rationale of the choice of using only PCR on blood for A. abstrusus

- what are the sensitivity and sensibility of PCR on blood for A. abstrusus

- how the authors intend to compare a blood-based PCR for A. abstrusus to the conventional PCR used in the vast majority of epidemiological studies, that is performed on faeces (Baermann sediment) or pharyngeal mucous for A. abstrusus

- if and how the authors standardized the method they used.

- how they explain the positivity to A. abstrusus at PCR of blood samples (and they should also reference this) 

In the method section, the authors indeed refer only to a paper on the molecular detection of Gurltia, but not of Aeluro. Indeed, the sensitivity/specificity of this technique in finding A. abstrusus DNA in the blood of cats is currently unknown.

Done, The semi-nested PCR method was selected because one of the study aims was to analyze animals infected with Gurltia paralysans that were additionally infected with aelurostrongilosis. The semi-nested PCR test selected for G. paralysans, which uses serum samples, additionally allows the detection of A. abstrusus DNA. Due to the availability of a large number of feline serum samples obtained from three Veterinary Hospitals and from three different regions of Southern Chile, a semi-nested PCR molecular technique was used to simultaneously detect G. paralysans and A. abstrusus. However, and as correctly stated by the referee, the A. abstrusus-molecular detection test needs further validation for but is validated for feline gurltiosis, and these characteristics were highlighted in the manuscript:

“…Recently, a semi-nested PCR was designed for G. paralysans identification and concurrent detection of A. abstrusus [2]. This molecular assay allows detection of the 28S ribosomal specific sequences of G. paralysans and A. abstrusus in serum samples, and could be used for the detection of simultaneous infections of feline gurltiosis and aelurostrongylosis [2,9]” (Introduction: Page 3, line 98-102).

“… Although not yet validated for feline aelurostrongylosis, the specificity of the molecular assay used here was confirmed by genetic sequencing, and the high homology of A. asbtrusus DNA samples with sequences available in the GenBank database. (Results: Page 7, line 260-262).

“…Although the semi-nested PCR technique used in our study has been validated for intra vitam diagnosis of G. paralysansusing serum samples, further validation is required for A. abstrusus detection [2,9]. However, positive A. abstrusus DNA samples in our analysis were confirmed with posterior sequencing analysis for the 28S rRNA gene. The molecular analysis used in this study could be an alternative method for the diagnosis of mixed feline infections with G. paralysans and A. abstrusus as previously suggested [9].” (Discussion, Page 8, line 287-299)

“..Although the semi-nested PCR used in this study requires further validation for detection of A. abstrusus with complementary fecal analysis (Baermann technique) from naturally infected cats, the specificity of the method was validated by the high homology of A. abstrusus DNA samples with available sequences in the GenBank database. As already stated, further work is required for validation of this molecular diagnostic tool for aelurostrongylosis detection in blood samples, using simultaneous Baermann funnel assays. (Discussion, Limitations of the study, Page 9, 338-343).

“Due to the availability of a large number of feline serum samples obtained from three Veterinary Hospitals and from three different regions of Southern Chile, a semi-nested PCR molecular technique was used to simultaneously detect G. paralysansand A. abstrusus. This technique allows the amplification of region 28 srDNA of G. paralysans and region 28s rRNA of A. abstrusus [2,9].” (Material and Methods, Page 10, lines 379-382).

This primary major concern should be addressed before rendering this paper publishable.

The authors should also include somewhere in the introduction a clear explanation of the rationale for the usage of this technique, other than pointing out advantages and disadvantages in the discussion section, otherwise their results cannot be considered for publication.

Done, in the discussion was added the following sentence “...Although the semi-nested PCR used in this study requires further validation for detection of A. abstrusus with complementary fecal analysis (Baermann technique) from naturally infected cats, the specificity of the method was validated by the high homology of A. abstrusus DNA samples with available sequences in the GenBank database. As already stated, further work is required for validation of this molecular diagnostic tool for aelurostrongylosis detection in blood samples, using simultaneous Baermann funnel assays. (Discussion, Limitations of the study, Page 9, 338-343).

“Due to the availability of a large number of feline serum samples obtained from three Veterinary Hospitals and from three different regions of Southern Chile, a semi-nested PCR molecular technique was used to simultaneously detect G. paralysansand A. abstrusus. This technique allows the amplification of region 28 srDNA of G. paralysans and region 28s rRNA of A. abstrusus [2,9].” (Material and Methods, Page 10, lines 379-382).

2) The abstract is 340 and should be drastically reduced as a maximum of 200 words is indicated for the abstract in the authors guidelines. 

Done, the abstract was reduced (212 words).

3) The manuscript requires an extensive and careful English grammar check, as well as a check of the large number of typos contained in it. Also, the manuscript is redundant and the language should be simplified

Done, the manuscript was revised and checked for english grammar and a certificate was issued

4) The reference list should be checked very carefully, as well as the matching text/references throughout the text, that seem to be completely wrong.

For instance, refs 8 and 9 are put in relation with sentences on Gurltia, while these refs refer actually to A. abstrusus. This makes the verification of the information extremely difficult. As far as I understood by reading the reference list: the refs in the text from 6 on, should be considered with 6 numbers more in the reference list. For instance, as far as I understood the n.8 in the reference list corresponds to nr. 14 in the text (I think).

In any case, regarding A. abstrusus the authors refer to fairly old articles, i.e. Traversa et al., 2008 for biological aspects and Elsheikha et al., 2016, of 13 (really old) and 5 years ago respectively.

Done, the references list was checked and revised.

It is here strongly recommended to cite updated literature here, as new knowledge has been gained in the last years and reviewed in other very recent papers published on Pathogens, e.g.

https://pubmed.ncbi.nlm.nih.gov/33401704/ (generalist/epidemiological/biological review on feline lungworms)

https://pubmed.ncbi.nlm.nih.gov/33920104/ (review on the clinical aspects)

Done, the literature was updated as recommended by the reviewer.

5) The discussion section is quite inconclusive, and it is only partially differentiate between the techniques used in the different studies cited when comparing prevalences. Moreover, discussions should be reformulated basing on the major issues raised on the methodology used. A complete and extended revision of English is strongly recommended, as well as a careful rephrasing as information is difficult to understand in the present form. Updated literature should be cited.

Done, the discussion added “...Although the semi-nested PCR used in this study requires further validation for detection of A. abstrusus with complementary fecal analysis (Baermann technique) from naturally infected cats, the specificity of the method was validated by the high homology of A. abstrusus DNA samples with available sequences in the GenBank database. As already stated, further work is required for validation of this molecular diagnostic tool for aelurostrongylosis detection in blood samples, using simultaneous Baermann funnel assays. (Discussion, Limitations of the study, Page 9, 338-343).

“Due to the availability of a large number of feline serum samples obtained from three Veterinary Hospitals and from three different regions of Southern Chile, a semi-nested PCR molecular technique was used to simultaneously detect G. paralysansand A. abstrusus. This technique allows the amplification of region 28 srDNA of G. paralysans and region 28s rRNA of A. abstrusus [2,9].” (Material and Methods, Page 10, lines 379-382).

MINOR COMMENTS

Lines 54-55: This sentence is redundant and it is not clear what the authors mean for “angio and neuotropic tropism”. Please rephrase

Done, the phrase was changed.

Done, it was changed to “This nematode resides mainly in the veins of the subarachnoid spinal cord and spinal cord parenchyma”

Line 57: replace “this parasite” with “G. paralysans

Done, it was changed

Lines 61-69: The authors should be careful when writing “the diagnosis of gurltiosis consists basically of”. Indeed, all this part should be preferabily presented as:

Clinical signs of feline gurltiosis

Laboratory alterations

Imaging diagnosis tools

Indeed, the authors themselves thereafter state that a diagnosis is achievable only post-mortem

Done, it was changed to “Clinical signs of feline gurltiosis normally include chronic or asymmetric ataxia of the pelvic limbs, ambulatory paraparesis, unilateral and bilateral hyperactive patellar reflexes, proprioceptive deficits, weight loss, coprostasis, and urinary and fecal incontinence. Laboratory alterations include mild levels of anemia, eosinophilia, and thrombocytopenia. Imaging findings by conventional myelography (CM) or computed tomographic myelography (CT-myelography), or magnetic resonance imaging (MRI), show thinning and obstruction of the dorsal and ventral spine in the thoracolumbar region, as well as a diffuse enlargement of the spinal cord in the thoracic, lumbar, and sacral regions [1,6,8]. (Introduction, Page 2, lines 63-70).

 Line 72: Please check “pre-and adult the adult”.

Done, it was changed to “pre-and adult nematodes”

Lines 73-74 It is not clear in which circumstances and which biological samples should be subjected to PCR for a diagnosis of G. paraylisans infections. Please specify4

Done, it was added “..PCR in serum”…

Line 83-85 Please cite that different species of gastropods are known to act as IH for A. abstrusus, as shown in both experimental and field studies:

https://pubmed.ncbi.nlm.nih.gov/31002674/

https://pubmed.ncbi.nlm.nih.gov/32650821/

https://www.mdpi.com/2076-0817/10/8/960

https://pubmed.ncbi.nlm.nih.gov/33322102/

Done, the articles were cited

Line 118-151 Scientific names are not in Italics here. Please change

Done, all the names were changed to italics.

Line 94 it is swabs not swaps. Please refer to updated literature on the diagnostic tools. Indeed, to the best of this reviewer’s knowledge PCR on blood samples has very little (if not nil) utility in supporting the diagnosis of feline aelurostrongylosis (major comments).

Done, it was added “…Molecular detection by PCR from feces or pharyngeal swabs can be used as alternative diagnostic tools for feline aelurostrongylosis [25,26]. An ELISA test for detection of A. abstrusus antibodies in serum has been used to improve the diagnosis of feline aelurostrongylosis [24, 27]. Recently, a semi-nested PCR was designed for G. paralysansidentification and concurrent detection of A. abstrusus [2]. This molecular assay allows detection of the 28S ribosomal specific sequences of G. paralysans and A. abstrusus in serum samples, and could be used for the detection of simultaneous infections of feline gurltiosis and aelurostrongylosis [2,9]. The presence of feline aelurostongylosis in South America has been reported in Uruguay, Argentina, Brazil, Colombia, Bolivia, and Chile [28,29].

Line 95-98 The authors propose a molecular based study, and this is quite innovative other than very useful for studies investigating lungworms in South America, as recently proposed in recent paper on lungworms in Brazil: https://pubmed.ncbi.nlm.nih.gov/34068219/

Line 108: questionnaire

Done

Line 255 maybe the authors meant “clinically healthy” cats

Done, it was changed

Line 260 “report.. have reported”. Please delete redundancies throughout the text

Done, it was changed

Line 270 suppose it “the” here and not “de”. It would be of benefit to provide explanations for the finding in Tenerife.. maybe is it due to increased animal movimentations or to a lack of awareness? Please specify and reference

Done, it was added “…The presence of G. paralysans in Tenerife Island could be due to the introduction of G. paralysans-infected domestic cats from endemic areas of South America, or the importation of either infected IHs or PHs [1,13].

Done, it was added “..The presence of G. paralysans in Tenerife Island could be due introduction of G. paralysans-infected domestic cats from endemic areas of South America or the importation of infected IH [1,13]. “

Line 285 determined

Done, it was changed

Line 301 the list of references regarding Europe should be updated. These are only examples of very recent studies:

https://pubmed.ncbi.nlm.nih.gov/30857861/

https://pubmed.ncbi.nlm.nih.gov/32748040/ (different techniques)

https://pubmed.ncbi.nlm.nih.gov/31991881/ (molecular study - faeces)

https://pubmed.ncbi.nlm.nih.gov/30552978/

Done, the articles were added.

The authors should explain why they decided to use a PCR on blood for the detection of A. abstrusus, since this method is not the best option for its diagnosis. Moreover, its sensitivity/specificity compared to best-known techniques (e.g. Baermann method, serology, PCR on other biological samples such as Baermann sediment or pharyngeal swabs - mucus) is unknown.

Done, as previously explained, it was added:

“…Recently, a semi-nested PCR was designed for G. paralysans identification and concurrent detection of A. abstrusus [2]. This molecular assay allows detection of the 28S ribosomal specific sequences of G. paralysans and A. abstrusus in serum samples, and could be used for the detection of simultaneous infections of feline gurltiosis and aelurostrongylosis [2,9]” (Introduction: Page 3, line 98-102).

“… Although not yet validated for feline aelurostrongylosis, the specificity of the molecular assay used here was confirmed by genetic sequencing, and the high homology of A. asbtrusus DNA samples with sequences available in the GenBank database. (Results: Page 7, line 260-262).

“…Although the semi-nested PCR technique used in our study has been validated for intra vitam diagnosis of G. paralysansusing serum samples, further validation is required for A. abstrusus detection [2,9]. However, positive A. abstrusus DNA samples in our analysis were confirmed with posterior sequencing analysis for the 28S rRNA gene. The molecular analysis used in this study could be an alternative method for the diagnosis of mixed feline infections with G. paralysans and A. abstrusus as previously suggested [9].” (Discussion, Page 8, line 287-299)

“..Although the semi-nested PCR used in this study requires further validation for detection of A. abstrusus with complementary fecal analysis (Baermann technique) from naturally infected cats, the specificity of the method was validated by the high homology of A. abstrusus DNA samples with available sequences in the GenBank database. As already stated, further work is required for validation of this molecular diagnostic tool for aelurostrongylosis detection in blood samples, using simultaneous Baermann funnel assays. (Discussion, Limitations of the study, Page 9, 338-343).

“Due to the availability of a large number of feline serum samples obtained from three Veterinary Hospitals and from three different regions of Southern Chile, a semi-nested PCR molecular technique was used to simultaneously detect G. paralysansand A. abstrusus. This technique allows the amplification of region 28 srDNA of G. paralysans and region 28s rRNA of A. abstrusus [2,9].” (Material and Methods, Page 10, lines 379-382).

Reviewer 2 Report

In this manuscript, the authors presented a molecular epidemiological study of the prevalence of Gurltia paralysans and Aelurostrongylus abstrusus in cats from Chile. The study included 171 cats from 3 cities in Southern Chile. The most important advantage of the survey is that it is the largest investigation of the prevalence of G. paralysans in cats. Furthermore, it is an important survey since this parasite is extremely poorly studied.

I have a few comments that should be corrected/considered:

  1. Please use a period instead of a comma in decimal numbers.
  2. Please italicize species names (including in keywords, table and figure captions, and references).
  3. Please correct the font size and style as required by the journal (especially text in tables and table captions).
  4. Figure 1 is incomplete - please correct it.
  5. Figure 2, Figure 3, and Table 5- It is not necessary to show these results. Please consider moving them to supplementary materials.
  6. Lines 143-144: ‘The results of the present study shown lack of association between analyzed potential risk factors and risk of infection for A. abstrusus and G. paralysans (Tables 3 and 4)’ - Is there no relationship between the use of anthelminthics and the occurrence of A. abstrusus?
  7. Please change the heading of section 4.3 – ‘Extraction and quantification of DNA from Gurltia paralysans
  8. Please correct the abbreviation of the weight unit from ηg to ng (lines 163, 400, and 424).
  9. Line 381-382: Please provide the country and city of manufacturer of the following reagents: OMEGA Bio-tek E.Z.N.A. kit, and Tissue DNA Kit D3396-02.
  10. Please correct the numbering of citations in the references section and in the main text (in the references section: numbers 1-6 have been duplicated, number 47 is missing).

Author Response

RESPONSE TO REVIEWER 2

In this manuscript, the authors presented a molecular epidemiological study of the prevalence of Gurltia paralysans and Aelurostrongylus abstrusus in cats from Chile. The study included 171 cats from 3 cities in Southern Chile. The most important advantage of the survey is that it is the largest investigation of the prevalence of G. paralysans in cats. Furthermore, it is an important survey since this parasite is extremely poorly studied.

I have a few comments that should be corrected/considered:

  1. Please use a period instead of a comma in decimal numbers.

Done, it was changed.

  1. Please italicize species names (including in keywords, table and figure captions, and references).

Done, it was changed

  1. Please correct the font size and style as required by the journal (especially text in tables and table captions).

Done, it was changed

  1. Figure 1 is incomplete - please correct it.

Done, I was corrected

  1. Figure 2, Figure 3, and Table 5- It is not necessary to show these results. Please consider moving them to supplementary materials.

Done, those were included as supplementary material

  1. Lines 143-144: ‘The results of the present study shown lack of association between analyzed potential risk factors and risk of infection for A. abstrusus and G. paralysans (Tables 3 and 4)’ - Is there no relationship between the use of anthelminthics and the occurrence of A. abstrusus?

Multivariable analysis shown no influence of antihelminthics and occurrence of A. abstrusus and/or G. paralysans.

  1. Please change the heading of section 4.3 – ‘Extraction and quantification of DNA from Gurltia paralysans

Done, it was changed to “Extraction and quantification of G. paralysans and A. abstrusus-specific DNA

  1. Please correct the abbreviation of the weight unit from ηg to ng (lines 163, 400, and 424).

Done, it was corrected

  1. Line 381-382: Please provide the country and city of manufacturer of the following reagents: OMEGA Bio-tek E.Z.N.A. kit, and Tissue DNA Kit D3396-02.

Done, it was added

  1. Please correct the numbering of citations in the references section and in the main text (in the references section: numbers 1-6 have been duplicated, number 47 is missing).

Done, the reference list was checked and revised.

Round 2

Reviewer 1 Report

Dear authors,

the work looks conceptually improved as you have specified the limitations of the technique used, that indeed is very interesting and innovative. I believe that something should be still checked and there is still the need to specify that this protocol should be compared with classical PCR protocols other than the Baermann (please find minor comments below). After addressing these suggestions, the ms could be considered suitable for publication

Please carefully check the text for English once again and for minor spell/check/typos (e.g. write correctly “abstrusus”)

Abstract

Please specify clearly that an experimental PCR protocol has been used.

Line 40 please delete the word “infections”

Lines 43, 150, 317 write correctly abstrusus.

Keywords: please write scientific names in italics

Line 61: please write: The adult stages of G. paralysans live in…

Line 66: it was fine to talk about the family Angiostrongylidae. Please revise according to NCBI taxonomy: https://www.ncbi.nlm.nih.gov/Taxonomy/Browser/wwwtax.cgi?id=1280927

Line 73: please use constipation instead of coprostasis

Line 94: please specify here that A. abstrusus  is also an angiostrongylidae

Line 112-116. Please specify here that this protocol is still experimental for A. abstrusus

Specify that this PCR is used on serum samples also here.

Line 145: infection was fine here

Line 150: please write “the percentage rate of infection” instead of levels

Line 289: on the presence

Line 295: please stop the sentence at previously. You can delete all the rest

Line 311: or to the importation

Line 315: maybe this part could be deleted as it has already been specified later in the revised version. Please, avoid these kind of repetitions.

Line 317: please check that the names are spelled correctly

Line 352-355: please do not use the term coprological, but rather substitute it with “copromicroscopic” here and elsewhere

Line 356 please specify on which biological sample was performed the nested PCR on false negative cats

Paragraph 4.3: It is not clear if whole blood or serum was subjected to the PCR. Please clarify

Lines 348-362 and Lines 409-415: It is this reviewer opinion that sentences from 356 to 362 could be integrated in a unique paragraph on the limitations along with lines 409-415.

Also, the authors should clearly state that this protocol is still experimental for A. abstrusus and that it should be compared in terms of sensitivity and specificity with classical PCR protocols, other than copromicroscopy (as correctly already done by the authors).

Line 451-454: It is not clear what the authors mean here.

Author Response

REVIEWER 1 – RESPONSE (Second round)

Dear authors,

the work looks conceptually improved as you have specified the limitations of the technique used, that indeed is very interesting and innovative. I believe that something should be still checked and there is still the need to specify that this protocol should be compared with classical PCR protocols other than the Baermann (please find minor comments below). After addressing these suggestions, the ms could be considered suitable for publication

Please carefully check the text for English once again and for minor spell/check/typos (e.g. write correctly “abstrusus”)

Done, it was revised.

Abstract

Please specify clearly that an experimental PCR protocol has been used.

Done, the phrase was added. “…were analyzed by an experimental semi-nested PCR protocol.”

Line 40 please delete the word “infections”

Done, it was delated

Lines 43, 150, 317 write correctly abstrusus.

Done, it was corrected

Keywords: please write scientific names in italics

Done, they were corrected. Apparently, during the transformation of the word document by the journal platform, some errors such as the change of the italic words occurred. These words were italicized in the initial previous versions of the manuscript.

Line 61: please write: The adult stages of G. paralysans live in…

Done, it was re-written.

Line 66: it was fine to talk about the family Angiostrongylidae. Please revise according to NCBI taxonomy: https://www.ncbi.nlm.nih.gov/Taxonomy/Browser/wwwtax.cgi?id=1280927

Done, it was corrected

Line 73: please use constipation instead of coprostasis

Done, it was changed.

Line 94: please specify here that A. abstrusus  is also an angiostrongylidae

Done, it was added.

Line 112-116. Please specify here that this protocol is still experimental for A. abstrusus

Done, it was added “Recently, a semi-nested PCR for serum samples was designed for G. paralysans identification and concurrent detection of A. abstrusus [2]. Nevertheless, this PCR protocol is still experimental for A. abstrusus.”

Specify that this PCR is used on serum samples also here.

Done, it was added “Recently, a semi-nested PCR for serum samples was designed for G. paralysans identification and concurrent detection of A. abstrusus [2]. Nevertheless, this PCR protocol is still experimental for A. abstrusus.”

Line 145: infection was fine here

Done, it was added

Line 150: please write “the percentage rate of infection” instead of levels

Done, it was added.

Line 289: on the presence

Done, it was changed

Line 295: please stop the sentence at previously. You can delete all the rest

Line 311: or to the importation

Done, it was changed

Line 315: maybe this part could be deleted as it has already been specified later in the revised version. Please, avoid these kind of repetitions.

Done, it was delated

Line 317: please check that the names are spelled correctly

Done, it was corrected

Line 352-355: please do not use the term coprological, but rather substitute it with “copromicroscopic” here and elsewhere

Done, it was changed

Line 356 please specify on which biological sample was performed the nested PCR on false negative cats

Done, It was added “…by nested-PCR on pharyngeal swabs”

Paragraph 4.3: It is not clear if whole blood or serum was subjected to the PCR. Please clarify

Done, it was serum.

Lines 348-362 and Lines 409-415: It is this reviewer opinion that sentences from 356 to 362 could be integrated in a unique paragraph on the limitations along with lines 409-415.

Done, the sentences were integrated in discussion.

Also, the authors should clearly state that this protocol is still experimental for A. abstrusus and that it should be compared in terms of sensitivity and specificity with classical PCR protocols, other than copromicroscopy (as correctly already done by the authors).

Done, it was added “Limitations of the study included a lack of copromicroscopic methods (i.e., Baermann´s technique, fecal smears and/or fecal flotations) for the detection of L1. Although the semi-nested PCR technique used in our study has been validated for intra vitam diagnosis of G. paralysans using serum samples, further validation is required for A. abstrususdetection [2,9]. However, positive A. abstrusus DNA samples in our analysis were confirmed with posterior sequencing analysis for the 28S rRNA gene. The molecular analysis used in this study could be an alternative method for the diagnosis of mixed feline infections with G. paralysans and A. abstrusus as previously suggested [9]. As already stated, further work is required for validation of this molecular diagnostic tool for aelurostrongylosis detection in blood samples, using simultaneous Baermann funnel assays and classical PCR protocols.

Line 451-454: It is not clear what the authors mean here.

Done, the phrase was delated
